https://doi.org/10.1038/s41467-019-10421-8　　**OPEN**

# Efficient base editing for multiple genes and loci in pigs using base editors

Jingke Xie[1,2,3,4,8], Weikai Ge[1,2,3,4,8], Nan Li[1,2,3,4,8], Qishuai Liu[1,2,3,4,8], Fangbing Chen[1,2,3,4], Xiaoyu Yang[1,3,4,5], Xingyun Huang[1,2,3,4], Zhen Ouyang[1,3,4], Quanjun Zhang[1,3,4], Yu Zhao[1,3,4], Zhaoming Liu[1,3,4], Shixue Gou[1,2,3,4], Han Wu[1,3,4], Chengdan Lai[1,3,4], Nana Fan[1,3,4], Qin Jin[1,2,3,4], Hui Shi[1,2,3,4], Yanhui Liang[1,2,3,4], Ting Lan[1,2,3,4], Longquan Quan[1,3,4], Xiaoping Li[1,3,4], Kepin Wang[1,3,4] & Liangxue Lai[1,3,4,6,7]

Cytosine base editors (CBEs) enable programmable C-to-T conversion without DNA double-stranded breaks and homology-directed repair in a variety of organisms, which exhibit great potential for agricultural and biomedical applications. However, all reported cases only involved C-to-T substitution at a single targeted genomic site. Whether C-to-T substitution is effective in multiple sites/loci has not been verified in large animals. Here, by using pigs, an important animal for agriculture and biomedicine, as the subjective animal, we showed that CBEs could efficiently induce C-to-T conversions at multiple sites/loci with the combination of three genes, including *DMD*, *TYR*, and *LMNA*, or *RAG1*, *RAG2*, and *IL2RG*, simultaneously, at the embryonic and cellular levels. CBEs also could disrupt genes (*pol* gene of porcine endogenous retrovirus) with dozens of copies by introducing multiple premature stop codons. With the CBEs, pigs carrying single gene or multiple gene point mutations were generated through embryo injection or nuclear transfer approach.

[1] CAS Key Laboratory of Regenerative Biology, Guangdong Provincial Key Laboratory of Stem Cell and Regenerative Medicine, South China Institute for Stem Cell Biology and Regenerative Medicine, Joint School of Life Sciences, Guangzhou Institutes of Biomedicine and Health, Chinese Academy of Sciences, Guangzhou Medical University, Guangzhou 510530, China. [2] University of Chinese Academy of Sciences, Beijing 100049, China. [3] Guangzhou Regenerative Medicine and Health Guangdong Laboratory (GRMH-GDL), Guangzhou 510005, China. [4] Institute for Stem Cell and Regeneration, Chinese Academy of Sciences, Beijing 100101, China. [5] Institute of Physical Science and Information Technology, Anhui University, Hefei 230601, China. [6] Jilin Provincial Key Laboratory of Animal Embryo Engineering, Institute of Zoonosis, College of Veterinary Medicine, Jilin University, Changchun 130062, China. [7] School of Biotechnology and Health Sciences, Wuyi University, Jiangmen 529020, China. [8] These authors contributed equally: Jingke Xie, Weikai Ge, Nan Li, Qishuai Liu. Correspondence and requests for materials should be addressed to K.W. (email: wang_kepin@gibh.ac.cn) or to L.L. (email: lai_liangxue@gibh.ac.cn)

Precise point mutation in the genome of an organism has great potential for agricultural and biomedical applications. Traditional methods precisely introduce or correct genetic point mutations by CRISPR-Cas9 and CRISPR-Cpf1 through the homology-directed repair (HDR) pathway requiring DNA double-strand breaks (DSBs) and exogenous donor DNA templates[1–4]. However, DSBs could create large deletion, complex genomic rearrangement, and targeted chromosome elimination when targeting several genomic loci simultaneously[5–7], which might lead to excessive DNA damage and cell death. In addition, HDR occurs infrequently (typically ~0.1–5%)[8], especially in non-dividing cells, thus impeding the use of HDR for precise genome editing in many species.

The newly developed base editors (BEs)[8–10] efficiently enable precise and highly predictable nucleotide substitutions (C-to-T or A-to-G conversion) at targeted genomic loci, independent of DSBs and donor templates. Thus, these BEs have great potential applications for both agriculture and biomedicine. To date, cytosine base editors (CBEs) have been successfully applied in plants (such as Arabidopsis, rice, wheat, and maize)[11,12], mammals (such as mice[13–16], rats[17], and rabbits[18]), and human cells[19–21] and embryos[22–24]. Many traits in agriculture are caused by multiple SNPs, and many genetic diseases in biomedicine arise from point mutations in multiple sites. Therefore, base editing of a genome in multiple sites is necessary to achieve favorable traits in agriculture, to establish human disease animal models, and to treat human hereditary diseases. However, all the reported cases only involved C-to-T substitution at a single targeted genomic site by using a single sgRNA. Whether C-to-T substitution is effective in base gene editing for multiple sites of a genome of large animals has not been verified.

In this study, we test the multiplexed base editing efficiency of BE3 (rAPOBEC1-XTEN-nCas9-UGI) and hA3A-BE3 (hAPOBEC3A-XTEN-nCas9-UGI) in pig, which is an important animal for both agriculture and biomedicine. We first confirm that the BE3 system was able to generate multiplexed base editing efficiently in embryonic and cellular levels. We also realize multiple copy base editing with a single sgRNA for porcine endogenous retrovirus (PERV) in porcine embryos and fibroblasts. Furthermore, with the CBE system, we achieve gene editing in pigs with single gene point mutation as well as multiple gene point mutations through either embryo injection or somatic cell nuclear transfer (SCNT) approach.

## Results

**Efficient base editing for multiple sites in porcine embryos**. To confirm whether base editing for multiple sites can be achieved by BE3 in the genome of porcine embryos (Fig. 1a), we selected six porcine genes, namely, *DMD*, *TYR*, *LMNA*, *RAG1*, *RAG2*, and *IL2RG*, which encode dystrophin, tyrosinase, lamin A/C, RAG1 protein, RAG2 protein, and IL-2 receptor gamma chain, respectively, as the target sites (Fig. 1b). Premature stop codons (Q493STOP and Q28STOP) would be generated by a single C-to-T conversion at the target sites in *DMD* and *TYR*, which are expected to result in Duchenne muscular dystrophy (DMD) and albinism, respectively. For *LMNA*, C-to-T conversion (a synonymous mutation, G608G) at the target site would create a cryptic splice donor site that produces a truncated splicing mutant of lamin A/C protein, termed "progerin", resulting in Hutchinson-Gilford Progeria Syndrome (HGPS)[25]. For *RAG1*, *RAG2*, and *IL2RG*, premature stop codons are expected to be generated due to C-to-T conversion. Mutations in *RAG1* or *RAG2* would result in the complete absence of B and T cells[26], and *IL2RG* mutations lead to the absence or profound depletion of T and natural killer (NK) cells without affecting the number of B cells[27]. *RAG1*-,

*RAG2*-, and *IL2RG*-deficient pigs would lack B cells, T cells, and NK cells (B/T/NK cells). In vitro transcribed DTL (*DMD*-sgRNA, *TYR*-sgRNA, and *LMNA*-sgRNA) or R12I (*RAG1*-sgRNA, *RAG2*-sgRNA, and *IL2RG*-sgRNA) with BE3 mRNAs (150 ng/μL) were co-injected together into 100 porcine parthenogenetically activated (PA) oocytes, which were conveniently available by in vitro matured oocytes derived from the ovary of slaughtered pigs. The injected PA embryos were cultured for 6 days post-parthenogenetic activation until blastocyst formation. The blastocyst rate of embryos injected with BE3 mRNA and DTL or R12I was 24% (24/100) and 25% (25/100) (Fig. 1e), respectively, which is consistent with the blastocyst rate of the embryos injected with water (24%, 24/100), indicating that BE3 and sgRNAs did not affect the early development of porcine embryo. A single blastocyst was lysed individually for genotyping by Sanger sequencing. Targeted point mutations were observed in 18 out of 24 (75%) embryos for DTL and 14 out of 22 (63.6%) embryos for R12I at target sites. Of these screened blastocysts, 12 for DTL (50%, 12/24) and 8 for R12I (36.4%, 8/22) were identified to have C-to-T mutations in three genes (Fig. 1c–e; Supplementary Fig. 1), four for DTL (16.7%, 4/24), and five for R12I (22.7%, 5/22) showed double-gene base editing (one for *DMD* and *TYR*, three for *DMD* and *LMNA*, one for *RAG1* and *RAG2*, two for *RAG1* and *IL2RG*, and two for *RAG2* and *IL2RG*), two for DTL (8.3%, 2/24) and one for R12I (4.5%, 1/22) showed single-gene base editing (one for *TYR*, one for *LMNA*, and one for *IL2RG*) (Fig. 1e). In addition to the specific conversion of C-to-T mutation, a few C-to-A substitutions were also found at expected positions of *TYR*-sgRNA (Fig. 1c; Supplementary Fig. 1a). In some embryos, C-to-T conversions occurred not only at positions 4–8 but also at position 13 for *TYR*-sgRNA, position 11 for *LMNA*-sgRNA, position 9 for *RAG1*-sgRNA, position 3 for *RAG2*-sgRNA, and positions 3, 9, and 10 for *IL2RG*-sgRNA, which would lead to the corresponding amino acid changes (S30L, S610F, S62F, S2L, V154V, L156L, and Q157STOP, respectively) (Fig. 1c, d; Supplementary Fig. 1a, b).

Recently, a new member of CBEs, hA3A-BE3, has been harnessed for base editing the genomes of mammalian cells and plants with improved efficiency and specificity[28–30]. We also tested multiple base editing capacity of hA3A-BE3 in pig embryos. As in BE3, DTL and R12I were used to evaluate the hA3A-BE3 system in porcine embryos. As shown in Supplementary Fig. 2a–e, 50% embryos for DTL (5/10) and 41.7% embryos for R12I (5/12) harbored C-to-T mutations at target sites of all three genes, which were comparable with those of the BE3 system (50% for DTL and 36.4% for R12I).

Some genes, such as *pol* gene of porcine endogenous retrovirus, which is a critical safety concern for xenotransplantation of pig organ to human, have multiple copies in the genome[31]. Induction of stop codons by BE3 could impede virus replication, possibly providing a new strategy to reduce PERV transmission. A sgRNA targeting the highly conserved catalytic center of the *pol* gene on PERVs was designed (Fig. 1f). Similarly, in vitro transcribed BE3 mRNA and *pol*-sgRNA were co-injected into the 100 PA embryos. At 6 days after in vitro culture, 20 blastocysts were harvested and genotyped by Sanger sequencing. As shown in Fig. 1g, h, and Supplementary Fig. 3a, b, all embryos harbored C-to-T conversions specifically at position 4 at frequencies ranging from 29.5% to 78.9%. Consequently, Q148STOP amino acid conversions occurred over all embryos. In addition, C-to-G conversion at position 4 was found in all embryos, with efficiencies ranging from 11.8% to 46.8% (Supplementary Fig. 3a, b). These results demonstrated that BE3 can simultaneously mediate the C-to-T conversion of several genes or a single gene with multiple copies with high efficiency in porcine embryos.

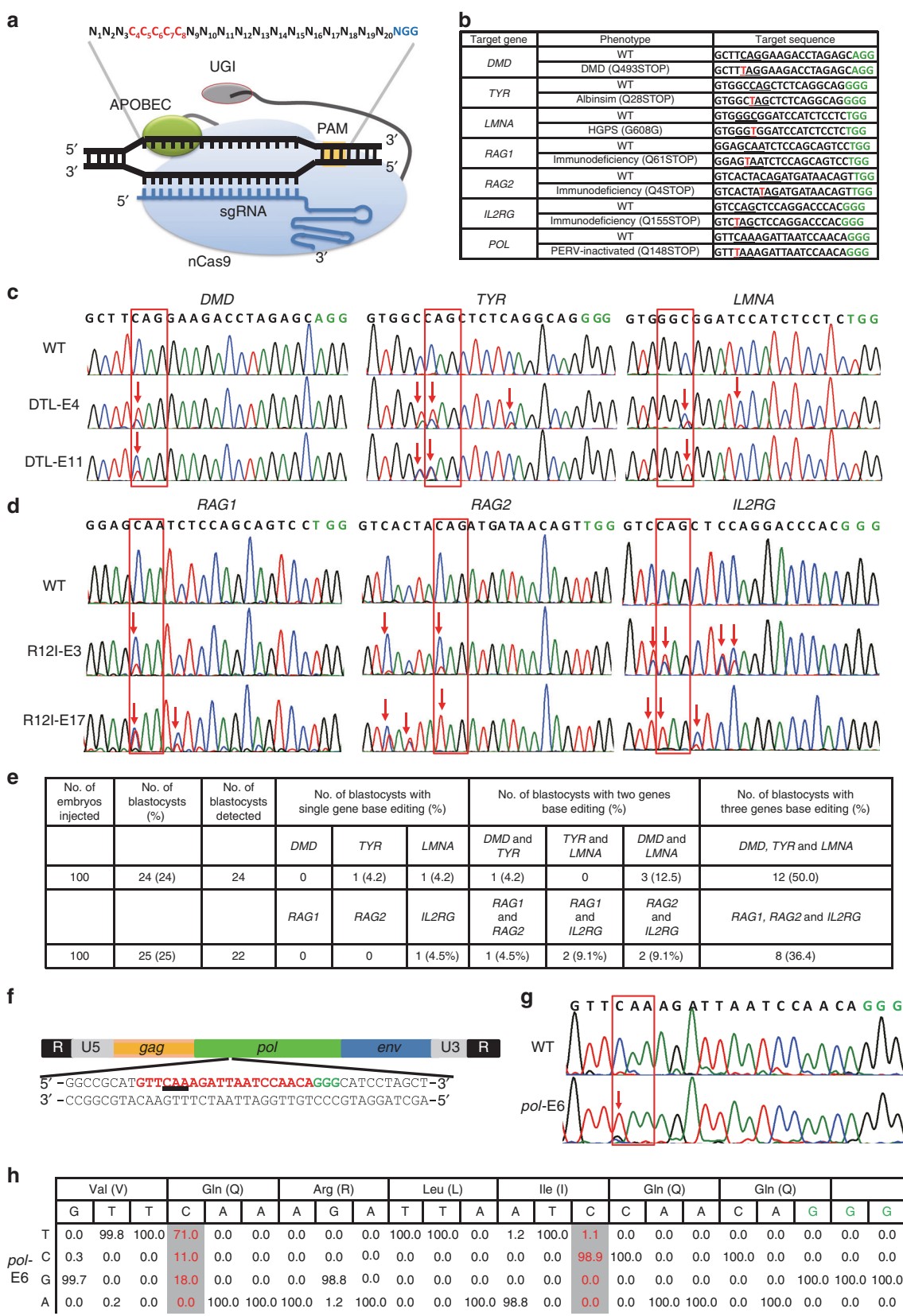

**Base editing at multiple loci in porcine somatic cells**. We next investigated whether BE3 could generate C-to-T conversions in multiple genes or multicopy genes simultaneously in porcine somatic cells. Six sgRNA candidates targeting exons of porcine *DMD*, *TYR*, *LMNA*, *RAG1*, *RAG2*, and *IL2RG* genes were synthesized and cloned into the BbsI-digested U6-sgRNA cloning vector (Fig. 1b). The mixed sgRNAs DTL (*DMD*-, *TYR*-, and *LMNA*-sgRNA) or R12I (*RAG1*-, *RAG2*-, and *IL2RG*-sgRNA)

**Fig. 1** The BE3 system can simultaneously induce C-to-T base editing at multiple genes/loci of porcine embryos. **a** Schematic of BE3-mediated C-to-T base editing. **b** Target-site sequences of *DMD*-, *TYR*-, *LMNA*-, *RAG1*-, *RAG2*-, *IL2RG*-, and *pol*-sgRNA. Target sequence (black), protospacer adjacent motif (PAM) region (green), target sites (red), and mutant amino acid (underlined). WT wild-type. **c** Sanger sequencing results of embryo-4# and 11# injected with *DMD*-sgRNA, *TYR*-sgRNA, and *LMNA*-sgRNA. The red box shows the successful C-to-T substitutions at target sites. **d** Sanger sequencing results of embryo-3# and 17# injected with *RAG1*-sgRNA, *RAG2*-sgRNA, and *IL2RG*-sgRNA. **e** Summary of multiple sites base editing by BE3 in porcine embryos. **f** Schematic of the PERV gene structure. One sgRNA targeting the catalytic region of the PERV pol gene was designed. The codon to be modified is underlined. The targeting sequence is in red and the PAM region is in green. **g** Representative sequence chromatogram of the target site of the *pol* gene from injected embryos 6#. Red box shows the successful C-to-T substitutions at target sites. **h** Nucleotide substitution frequencies mediated by BE3 and *pol*-sgRNA were measured in the injected embryos 6# by targeted deep sequencing

were co-transfected with BE3-expressing vectors into the porcine fetal fibroblasts (PFFs), which were then cultured in 10-cm dishes at a seeding density of 5 cells/cm² to form single-cell colonies. After 10–14 days of G418 selection, single-cell colonies were obtained and genotyped by Sanger sequencing. Sequencing results are summarized in Supplementary Data 1 and 2. For DTL, 26 single-cell colonies (25.2%, 26/103) showed C-to-T substitution of all three genes (Fig. 2a; Supplementary Fig. 4a), 18 (17.5%, 18/103, three for *DMD* and *TYR*, six for *TYR* and *LMNA*, and nine for *DMD* and *LMNA*) showed double-gene base editing, and 12 (seven for *DMD*, three for *TYR*, and two2 for *LMNA*) showed single-gene base editing (Fig. 2c). For R12I, three single-cell colonies (1.5%, 3/204) with C-to-T substitution of all three genes (Fig. 2b; Supplementary Fig. 4b), 18 (8.8%, 18/204, 15 for *RAG1* and *RAG2*, 1 for *RAG1* and *IL2RG*, and 2 for *RAG2* and *IL2RG*) with double-gene base editing, and 56 (12 for *RAG1* and 44 for *RAG2*) with single-gene base editing were identified (Fig. 2c). Specific C-to-T substitutions occurred within the canonical BE3 editing window (positions 4–8) as expected. As shown in Fig. 2b and Supplementary Fig. 4b, unwanted C-to-T substitutions at position 3 for *RAG2*-sgRNA and positions 3, 9, and 10 for *IL2RG*-sgRNA were observed, leading to the corresponding amino acid changes (S2L, V154V, L156L, and Q157STOP, respectively). Furthermore, unwanted C-to-A substitutions at position 8 for *RAG2*-sgRNA were observed in cell colony R12I-C22, resulting in an aminoacid change (Q4K). Thus, BE3 can efficiently induce C-to-T substitution in three different genes of porcine somatic cells in one step. We also tested the multiple base editing capacity of hA3A-BE3 in porcine cell level. As in BE3, DTL and R12I were used for evaluation of hA3A-BE3 in porcine fibroblasts. Base editing at target sites of all three genes occurred in the genomes of 55% single-cell colonies for DTL (11/20) and 40% for R12I (8/40), which was higher than BE3-mediated multiplex base editing (25.2% for DTL and 1.5% for R12I) (Supplementary Fig. 5a–e).

The BE3- and *pol*-sgRNA-expressing vectors were co-transfected into Bama-PFFs to induce the same premature stop codon into multiple copies of *pol* gene. Previous reports identified 25 copies of functional PERVs in the Bama-mini pigs[32]. In this study, we analyzed 155 single-cell-derived colonies by PCR and Sanger sequencing to confirm the C-to-T conversion. Sequencing results showed that 59 colonies (38.1%, 59/155) were confirmed to have C-to-T base editing at position 4 of *pol*-sgRNA. The copy number of PERV in 40 colonies (25.8%, 40/155) was reduced by more than half (Supplementary Fig. 6c). The 30# and 87# cell colonies (Supplementary Fig. 6a) were further subjected to high-throughput deep sequencing, and the frequencies of base substitutions were calculated. The C-to-T conversions occurred on 84.9% of targeted Colonies for 30# colony and 84.0% for 87# colony (Fig. 2d–f; Supplementary Fig. 6b), indicating that the copy number of PERV reduced by around 85% in both cell colonies. In addition, C-to-G substitutions were observed in 30# and 87# colonies with efficiencies of 2.0% and 2.4%, respectively (Fig. 2d–f; Supplementary Fig. 6b).

**Generation of *LMNA*^G608G^ pig via zygote injection**. Although single-cell colonies harboring desired *LMNA*^G608G^ mutation were effectively obtained (41.7%, 43/103) when using BE3 and *LMNA*-sgRNA (Fig. 2c), these colonies exhibited premature senescence phenotypes and were unsuitable for use as nuclear donor for SCNT (Supplementary Fig. 7). Therefore, we investigated whether the *LMNA*^G608G^ mutation pig model could be generated by using embryo injection of base editors. In vitro-transcribed BE3 mRNA and *LMNA*-sgRNA were co-injected into the cytoplasm of one-cell stage zygotes from four Large White sows. Forty-six injected porcine zygotes were then transferred into two surrogate Large White sows. A total of nine piglets (five male and four female, one stillborn and eight liveborn) were born at term from one pregnant surrogate (Fig. 3a, b). Genotype analysis showed that eight piglets (88.9%, 8/9) harbored C-to-T mutation (Fig. 3c, d; Supplementary Fig. 8). For 357-2, 357-3, 357-4, and 357-7 (44.4%, 4/9), undesired indels (1 bp or 12 bp deletion) were also found at target sites with the efficiency of 44%, 15%, 3%, and 28%, respectively (Fig. 3c; Supplementary Fig. 8). Targeted deep sequencing results showed that 13%-100% site-specific C-to-T mutations were achieved at *LMNA* c.1824C site for piglet 357-1, 357-2, 357-3, 357-5, 357-6, 357-7, 357-8, and 357-9 (Fig. 3c–e). Notably, piglet 357-8 harbored homozygous c.1824C-to-T mutations, but unfortunately died within 2 days (Fig. 3c–e). The heart, liver, spleen, lung, and kidney of piglet 357-8 were collected, and Sanger sequencing results showed that homozygous c.1824C-to-T mutations were observed in all these tissues (Supplementary Fig. 9). These results showed that pig models carrying C-to-T substitutions can be generated efficiently by direct injection of zygotes with BE3 system.

Examining the potential off-target (POT) effects is important to evaluate a new genome-editing tool. We computationally predicted POT sites using Cas-OFFinder (http://www.rgenome.net/cas-offinder/)[33]. Sanger sequencing analysis of seven POT sites showed that one off-target mutation (OT3) was found in eight (88.9%, 8/9) base-edited piglets (Supplementary Fig. 10), which are consistent with recent reports showing that CBEs can induce genome-wide off-target mutations in mammals[34] and plants[35].

To test whether *LMNA*^G608G^ mutation could cause aberrant mRNA splicing, total RNAs from the ear tissues were extracted. RT-PCR and Sanger sequencing analysis showed that ear tissues from all *LMNA*^G608G^ piglets expressed a smaller mRNA with a 150-nucleotide deletion (Fig. 3f, g). Western blot analysis of the heart, liver, spleen, lung, and kidney tissues of piglet 357-8 demonstrated the presence of progerin, a truncated splicing mutant of lamin A, in the tissues of *LMNA*^G608G^ mutation piglets but not in wild-type (WT) tissues (Fig. 3h). These results demonstrated that *LMNA*^G608G^ mutation piglets, consistent with human HGPS patients, can express progerin mRNA and protein.

**Generation of base editing pigs via SCNT**. *DMD* heterozygous female pig could grow normally. Mating *DMD*^−/+^ female pigs with *DMD*^+/Y^ could generate *DMD* knockout pig model with

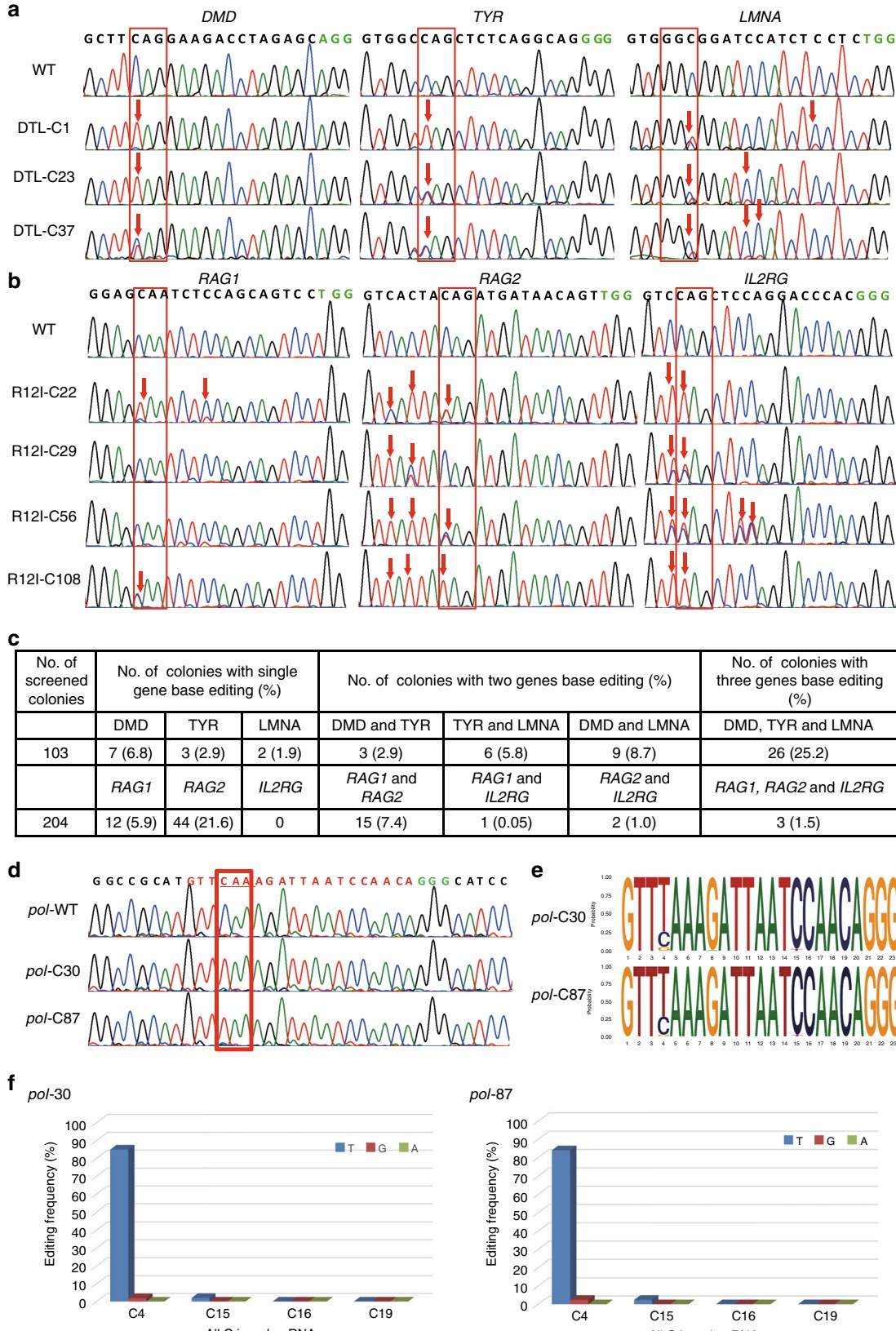

**Fig. 2** BE3-mediated base editing for multiple genes/loci in porcine somatic cells. **a**, **b** Sanger sequencing chromatograms of selected single-cell colonies. The codon in the red box indicate expected substitutions at target sites. **c** Summary of base editing in porcine fibroblasts. **d** Sanger sequencing results of selected single-cell colonies in the *pol* gene. **e**, **f** Summary of the targeted deep sequencing of on-target site for the *pol* gene (cell colonies 30# and 87#)

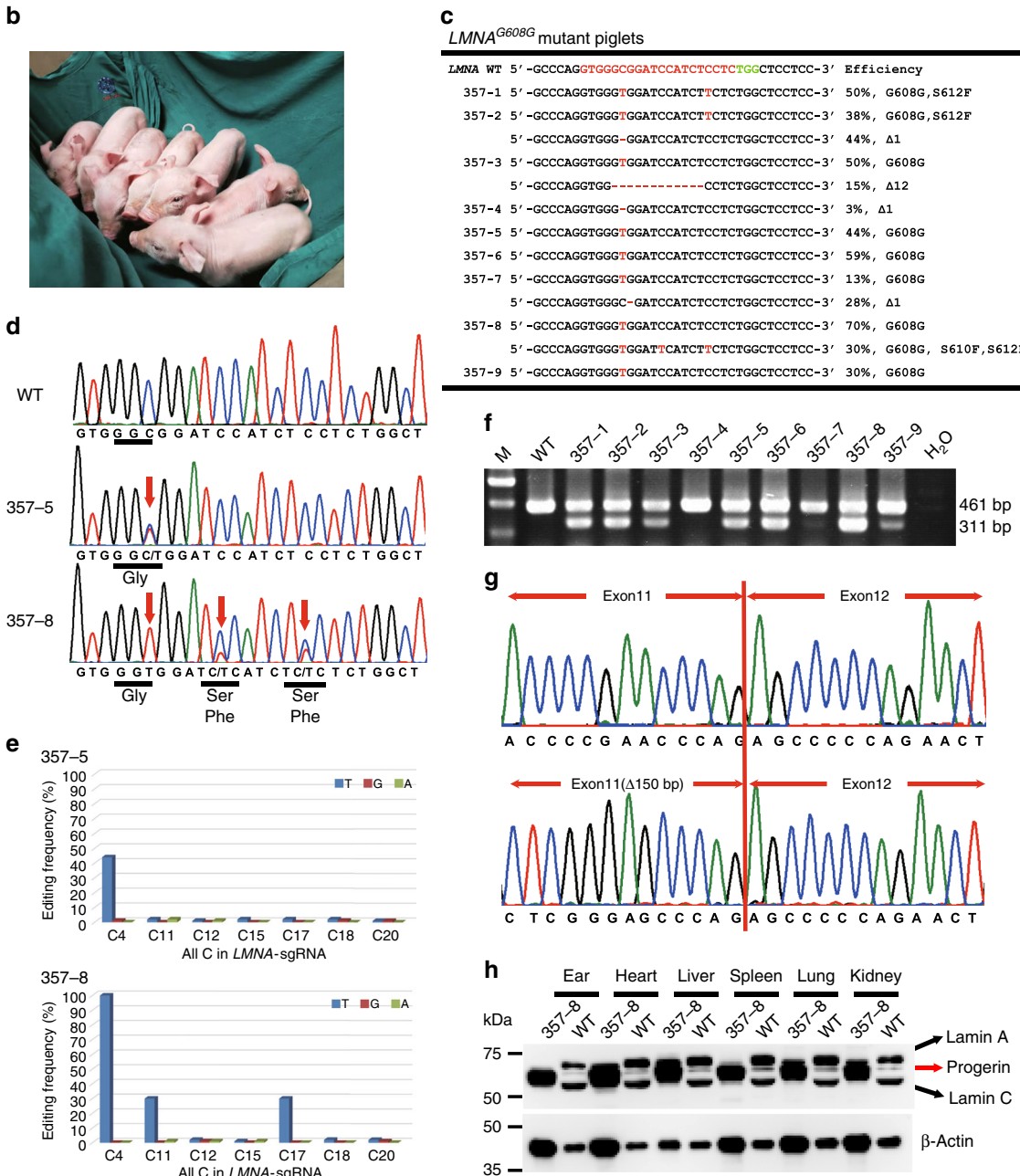

**Fig. 3** Generation of *LMNA^G608G* pig via direct zygote injection. **a** Summary of generation of *LMNA^G608G* mutant pigs by using direct zygote injection of the BE3 system. **b** Representative photograph of newborn *LMNA^G608G* piglets. **c** Summary of genotypes of nine newborn piglets from targeted deep sequencing. C-to-T substitutions and indels are shown in red. **d** Sanger sequencing chromatograms of WT, 357-5, and 357-8 piglets. The red arrow indicates the target sites with C-to-T conversions. **e** The efficiencies of C-to-T and non C-to-T substitutions in all Cs in LMNA-sgRNA were detected by targeted deep sequencing. **f** The expression of WT and truncated *LMNA* was detected by RT-PCR. Truncated *LMNA* mRNA is translated to progerin, which can result in HGPS. **g** Sanger sequencing chromatograms of RT-PCR products of WT and *LMNA^G608G* piglets. **h** Western blot was used to detect the expression of lamin A/C and progerin protein in the heart, liver, spleen, lung, kidney, and ear tissues of WT and 357-8 piglets. Source data are provided as a Source Data file

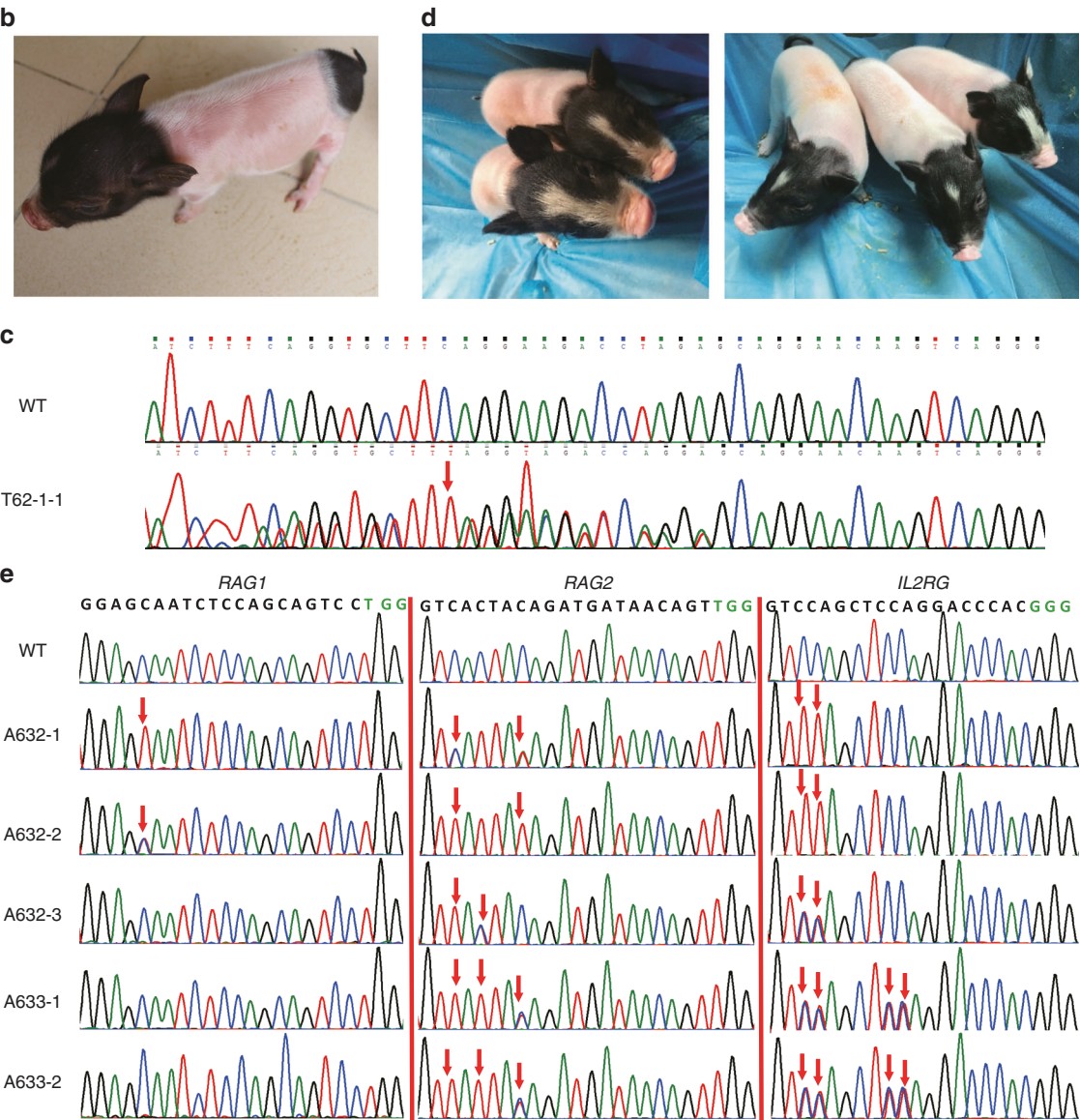

| Target genes | Cell pools | Transferred embryos | No. recipients | No. (%) pregnancies | No. cloned piglets | No. (%) piglets carrying mutations |
|---|---|---|---|---|---|---|
| *DMD* | C3, C17, C36, C59, C67, C87, C92 | 400 | 2 (T62-1, 51-1) | 1 (T62-1, 50%) | 1 (T62-1-1) | 1 (T62-1-1, 100%) |
| *RAG1, RAG2 and IL2RG* | C22, C29, C56, C108 | 1338 | 6 (A624, A625, A630, A631, A632, A633) | 2 (A632, A633, 33.3%) | 5 (A632-1, A632-2, A632-3, A633-1, A633-2) | 5 (A632-1, A632-2, A632-3, A633-1, A633-2, 100%) |

**Fig. 4** Base editing pigs were generated by SCNT. **a** Summary of SCNT results for generation of *DMD* mutant and *RAG1, RAG2,* and *IL2RG* mutant cloned pigs. **b** Image of *DMD* heterozygous female pigs produced in this study. **c** Genotype of *DMD* heterozygous female piglets. **d** Representative images of five cloned piglets with C-to-T conversions in *RAG1, RAG2,* and *IL2RG* genes. **e** Sanger sequencing chromatograms of DNA from wild-type and five cloned piglets. The red arrow indicates the substituted nucleotides

symptoms of patients[1]. For this purpose, seven cell colonies (*DMD*-C3, 17, 36, 59, 67, 87, and 92) with C-to-T conversion in one allele and wild-type or 18-bp deletion in the other allele were selected as nuclear donors for SCNT (Supplementary Fig. 11). A total of 400 cloned embryos were constructed and surgically transferred into two surrogates (T62-1, 51-1), and one surrogate (T62-1) was pregnant (Fig. 4a). After ~114 days of gestation, one live piglet (T62-1-1) was born (Fig. 4b). We amplified the *DMD*-sgRNA-targeted site from the genomic DNA of a newborn piglet by PCR, and then obtained Sanger-sequenced PCR products. Sequencing results showed that the cloned piglet harbored a C-to-

T mutation in one allele and 18-bp deletion in the other allele (Fig. 4c; Supplementary Fig. 12a, b). The cloned piglet grew healthy, and is more than 1-year old now.

Four single-cell colonies (R12I-C22, C29, C56, and C108) with *RAG1, RAG2,* and *IL2RG* mutations were selected for SCNT to confirm whether live pigs could be achieved through multiple base editing (Fig. 2b; Supplementary Fig. 4b). A total of 1338 reconstructed embryos were transferred into six surrogate pigs (A624, A625, A630, A631, A632, and A633). Of these surrogate animals, two became pregnant (A632, A633), yielding two males (A632-1 and A632-2) and three females (A632-3, A633-1, and

A633-2) piglets that were naturally delivered after 114 days of gestation (Fig. 4a, d). Genotype analysis showed that A632-1 contained CAA > TAA mutations in both alleles of the *RAG1* gene, CAG > TAG mutation in one allele and CAG > AAG mutation in the other allele of the *RAG2* gene, and CAG > TAG in both alleles of the *IL2RG* gene, which is consistent with the genotype of R12I-C22 cell colony. For A632-2 piglets, a premature stop codon was generated in one allele of the *RAG1* gene, while complete CAG > TAG conversions were achieved for the *RAG2* and *IL2RG* genes. This genotype was the same as that in R12I-C108 cell colony. The genotype of A633-1 and A633-2 piglets was CAG > TAG mutation in only one allele of *RAG2* and *IL2RG* genes, and no mutation was found for the *RAG1* gene. These two piglets originated from the SCNT of R12I-C59 cell colony. For A632-3, only a premature stop codon was achieved in one allele of the *IL2RG* gene, corresponding to the genotype of donor R12I-C29 cell colony (Fig. 4e; Supplementary Fig. 13).

We identified 21, 7, 8, and 10 POT sites with 2–3 nucleotide mismatches for *DMD*-sgRNA, *RAG1*-sgRNA, *RAG2*-sgRNA, and *IL2RG*-sgRNA, respectively. Sanger sequencing analysis of all PCR products indicated that no base substitution was detected at any of these POT sites of all cloned piglets (Supplementary Figs 14, 15a–c).

All cloned *RAG1*, *RAG2*, and *IL2RG* mutated piglets were raised in the conventional housing environment and piglets with immunodeficiency presented health issues. Consequently, piglets carrying homozygous *RAG1*, *RAG2*, or *IL2RG* mutations (A632-2, A632-1), which looked normal and strong upon birth, died 12 and 49 days after birth, respectively, due to infection in the lung (Fig. 5a; Supplementary Fig. 16). Cloned piglets carrying heterozygous *RAG2* and *IL2RG* mutations also died 41 and 75 days after birth due to the same reason. By contrast, piglets with only *IL2RG* heterozygous mutations (A632-3) grew normally and survived until they reached sexual maturity. Dead *RAG1*, *RAG2*, and *IL2RG* mutated piglets and killed age-matched WT piglets were autopsied immediately. The results showed that mutated piglets had undetectable or severely hypoplastic thymuses compared with age-matched wild-type piglets (Fig. 5b). Moreover, the spleens of the A632-1, A632-2, A633-1, and A633-2 piglets were smaller and thinner than those of the WT piglets (Fig. 5c). The thymuses and spleens were fixed and then subjected to hematoxylin–eosin (H&E) staining. In the thymuses of mutated piglets, the thymic lobules were atrophied and very few medulla-like areas were detected, whereas thymic lobules were clearly found in age-matched WT piglets (Fig. 5d). The white pulp was almost undetectable and showed hypoplastic lymphoid aggregations in the spleens of the mutated piglets, but not in the spleens of the WT piglets (Fig. 5e).

We detected whether *RAG1/2* and *ILR2G* knockout pigs lack mature B, T, and NK lymphocytes. Cells from the peripheral blood, spleen, and bone marrow were collected. Quantitative RT-PCR (qRT-PCR) analysis showed that *IL2RG*, T-cell surface markers *CD4* and *CD8*, and B-cell surface marker *CD19* were drastically reduced in the peripheral blood of two $IL2RG^{-/Y}$ (A632-1 and A632-2) and two $IL2RG^{+/-}$ (A633-1 and A633-2) piglets, but no expression changes were found in the A632-3 piglet (Fig. 6a). This result may be caused by the X chromosome inactivation of the *IL2RG* gene in female mammals. Sanger sequencing of the RT-PCR products of *IL2RG* showed that only mutant *IL2RG* was expressed in the A633-1 and A633-2 piglets, which caused spleen and thymus hypoplasia and immunodeficiency (Supplementary Fig. 17). The collected peripheral blood cells were then subjected to flow-cytometric analysis. After staining with CD3 antibodies, almost no CD3+ populations were detected in the peripheral blood of A632-1, A632-2, A633-1, and A633-2 (2.97%, 1.17%, 3.53, and 2.75%, respectively), confirming

the lack of mature T cells (Fig. 6b). However, ~31.2% and 55.9% CD3+ cells were harbored in the corresponding WT and A6332-3 piglets. Furthermore, IgM expression was detected to determine the status of B cells by FACS. IgM+ cells were almost undetectable in the PB of A632-1, A633-1, and A633-2 (0, 0.13%, and 0.01%, respectively), whereas the ratios of ~8.32% and 3.30% were observed in the WT and A632-3 counterparts, respectively (Fig. 6c). After staining with CD3 and CD8 antibodies, we almost failed to detect CD3+CD8+ cells in A632-1, A632-2, A633-1, and A633-2 (0.95%, 0.50%, 0.34%, and 0.63%, respectively), but the ratios of 2.39% and 7.67% were found in WT and A632-3 piglets, respectively. These results indicated the absence of NK cells in our obtained immunodeficient piglets (Fig. 6d).

V(D)J rearrangement is the unique mechanism of genetic recombination that occurs only in developing lymphocytes during the early stages of T- and B-cell maturation. We further analyzed whether or not V(D)J rearrangements of TCR and BCR had been blocked by the disruption of *RAG1* and *RAG2* genes. V (D)J rearrangements in the T-cell receptor β chain (TCRβ) locus were detected in the thymus and PB by PCR. PCR results showed no rearrangements at the TCRβ locus in the thymus and PB of the $RAG1^{-/-}$ (A632-1) and $RAG2^{-/-}$ (A632-2) piglets (Fig. 6e, f). For BCR, we selected the IgH locus to detect V(D)J rearrangements in the spleen, peripheral blood, and bone marrow. Similarly, the V(D)J rearrangements of the IgH locus were not detected in the *RAG1/2* knockout pigs, By contrast, V (D)J rearrangements at the TCRβ and IgH loci were observed in the WT control and A632-2 piglet (Fig. 6g–i). For the A633-1 and A633-2 piglets, V(D)J rearrangements dramatically decreased and became almost undetectable in the thymus, which may be caused by abnormal lymphocyte development due to *IL2RG* mutation.

## Discussion

BE3-mediated C-to-T substitution in the codons of open-reading frames can potentially cause amino acid substitutions (missense mutations), aberrant splicing or conversion of CAA, CAG, CGA, and TGG codons into stop codons (nonsense mutations)[36,37]. Induction of stop codons, the most frequent events resulting from C-to-T substitutions, as well as aberrant splicing, can induce degradation of mRNA transcripts by nonsense-mediated decay or the synthesis of truncated proteins, thus enabling gene disruption. In this work, among the seven genes tested, six genes, namely, *DMD*, *TYR*, *RAG1*, *RAG2*, *IL2RG*, and *pol*, could be induced a premature stop codon. By contrast, induction of an aberrant splicing mutation was the better choice for the other gene (*LMNA*).

Gene editing in animals could be realized d by co-injection of Cas9 mRNA and sgRNA into one-cell stage embryos. Therefore, we first tested efficiency of the BE3 system and hA3A-BE3 system for multiple base editing in embryo level. A combination of three functionally unrelated genes (*DMD*, *TYR*, and *LMNA*), and a combination of three immune-related genes (*RAG1*, *RAG2*, and *IL2RG*) were used for the respective experiments. Our results showed that, in embryo level, both BE3 system and hA3A-BE3 system were able to generate multiple base editing with high efficiency. Approximately 40–50% embryos contained C-to-T conversions in three genes simultaneously. The other blastocysts could contain cells with one or two gene mutations in some embryos. C-to-T conversions occurred not only at positions 4–8 but also at other positions, such as 9, 10, 11, and 13, which would lead to unwanted corresponding amino acid changes. In addition to the specific conversion of C-to-T mutation, undesired mutations such as C-to-A substitutions (*TYR*-sgRNA) and indels (1 - bp or 12 -bp deletion for *LMNA* gene, and 18 -bp deletion for

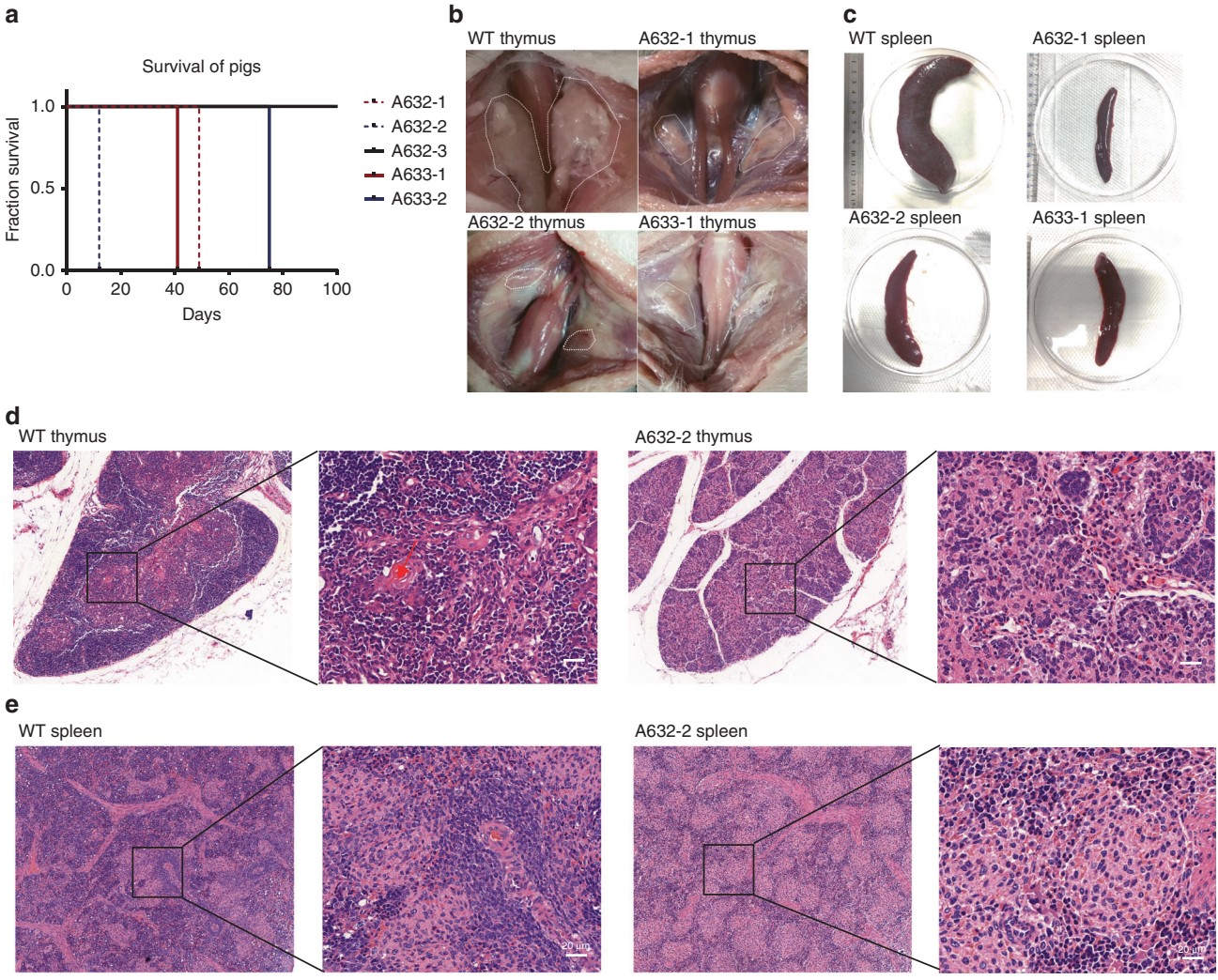

**Fig. 5** Phenotype of *RAG1*, *RAG2*, and *IL2RG* mutant piglets. **a** Survival curve for the five cloned piglets. **b**, **c** The hymus (**b**) and spleen (**c**) of piglets with base editing at *RAG1*, *RAG2*, and *IL2RG* genes were evidently smaller than those of age-matched WT piglets. **d**, **e** H&E staining results of the thymus (**d**) and spleen (**e**) of A632-2 and age-matched WT piglets. Scale bar, 20 μm

*DMD* gene) were also found at expected positions. These unwanted mutations had been also reported in other organisms and probably were caused by deamination and subsequent base-excision repair during the C-to-T conversion[21].

When using embryo injection to generate gene editing animals, the event of C-to-T conversion may happen in a cell of two or more cell-stage embryos. Thus, many of the resulting founder animals were chimeric ones mixed with mutant and non-mutant tissues, or mixed with homozygous cells and heterozygous cells. To acquire the animals with a pure pattern of mutation, one or two more rounds of further breeding have to be employed for selection among the offspring, which is expensive and time consuming. For multiple base editing, occurrence of the undesired mutations in founders would increase the complexity of the chimeric issues. Therefore, embryo injection is unsuitable for generating multiple-base editing large animals, such as pigs, with long gestation term and sex maturation time. However, when generating a single-base editing animal, for some genes, for instance, (e.g., *LMNA* gene, an aging-related gene, with special functions), embryo injection could be a better choice. Mutations of *LMNA* gene could cause premature aging syndromes. When we putatively established porcine fibroblast line-harboring *LMNA*^*G608G* mutation, the cells also exhibited premature senescence phenotypes in vitro, and were unsuitable for use as nuclear

donors for SCNT. Therefore, we applied embryo injection of base editors to establish a *LMNA*^*G608G* mutation pig model. HGPS pig models carrying a c.1824C-to-T point mutation were successfully generated via zygotes injection, indicating that pig models carrying C-to-T substitutions can be generated by direct zygote injection of BE3, thus providing an appropriate model for pre-clinical study of aging.

Gene targeting of large animals can be achieved through combination of gene targeting of somatic cells with somatic cell nuclear transfer[1,2,38]. In vitro screening of gene-edited fibroblasts with desired mutations prior to SCNT could circumvent several problems, such as chimerism and unwanted mutations, which are commonly found in embryo injection approach. In the proposed approach, given that the pig cloning technology has been well established, the most elusive issue for the generation of gene editing animal is to create mutant cells that can be used as donor nuclei. Therefore, we tested the multiple base editing efficiency of BE3 and hA3A-BE3 in porcine somatic cells. Our results showed that, both BE3 and hA3A-BE3 could be applied for efficient base editing in three genes simultaneously. Unlike that in embryo level, in which hA3A-BE3 and BE3 had similar efficiency, the efficiency of hA3A-BE3-mediated multiplex base editing was higher than that of BE3 in porcine cell level. Although as in the embryo level, BE3 and hA3A-BE3 systems could also result in cells with one or two gene

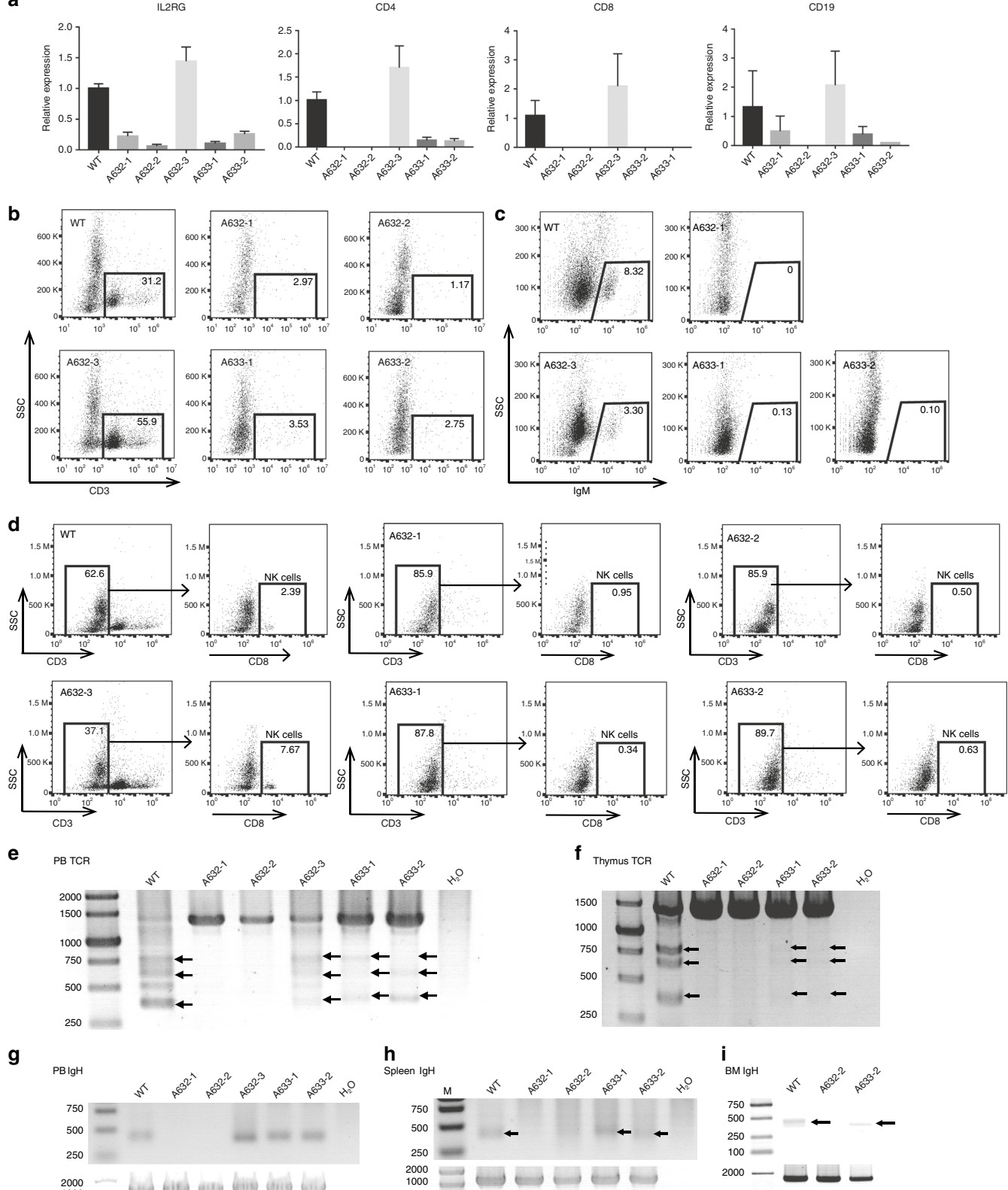

**Fig. 6** Immunocytes and V(D)J rearrangement analysis of *RAG1*, *RAG2*, and *IL2RG* mutant piglets. **a** qRT-PCR analysis of *IL2RG*, *CD4*, *CD8*, and *CD19* expression in peripheral blood of *RAG1*, *RAG2*, and *IL2RG* mutant piglets. **b–d** FACS analysis of mature B (**b**), T (**c**), NK (**d**) cells from the peripheral blood. Cells were stained with antibodies of anti-CD3, IgM CD8 to detect mature B (**b**), T (**c**), and NK (**d**) cells. **e**, **f** TCR-β gene rearrangement analysis in the peripheral blood and thymus. **g**, **h**, and **i** IgH gene rearrangement analysis in the peripheral blood (**g**), spleen (**h**), and bone marrow (**i**). Source data are provided as a Source Data file

mutations and unwanted mutations, such as undesired non-C-to-T substitutions and indels in some cell colonies. The cell lines with desired mutations can then be selected by genotyping to obtain cloned pigs with favorable phenotypes. The effectiveness of this approach was validated by the generation of pigs with point mutations in one and three genes. We first made a DMD pig model with C-to-T conversion of a single gene (*DMD* gene) through the BE3 system combining with SCNT approach. DMD is an X-linked recessive hereditary disease, and the average life expectancy of patients with this disorder is 26 years old[39]. Previously reported $DMD^{-/-}$ or $DMD^{-/Y}$ pigs made by deletion of a fragment through the CRISPR-Cpf1 system could not survive beyond 3 months[1]. Thus, we chose female cell colonies with $DMD^{-/+}$ mutation of C-to-T conversion to generate *DMD* heterozygous female pigs, which could survive more than 1 year and could be used to mate with $DMD^{+/Y}$ pigs to achieve many F1 pig models with DMD phenotypes for biomedicine study. The application of the BE3 system in generation of multiple base editing animal was validated by choosing the mutant cells with C-to-T conversions of three functionally related genes *RAG1*, *RAG2*, and *IL2RG* for SCNT. We successfully achieved immunodeficient pigs lacking B cells, T cells, and NK cells, which are consistent with the SCID pigs made by deletion of fragments with other gene editing technologies[40–42].

The risk of cross-species transmission of PERVs has impeded the clinical application of xenotransplantation from pig human. Previously, PERVs are inactivated in a porcine primary cell line by using CRISPR-Cas9[31,32]. Here, we proved that induction of stop codons can also disrupt genes with dozens of copies of *pol* gene on PERVs in the porcine genome. CRISPR-Cas9-mediated DSBs are deleterious for cell growth. To support clonal expansion of 100% PERV-inactivated cells, p53 inhibitor is added to mitigate the stress from multiplex DNA damage during multiplexable genome engineering. By using the BE3 system, instead of DSBs, stop codons were generated, thereby avoiding DNA damage and maintaining normal growth of mutant cells, thus, potentially facilitating disruption genes with multiple copies. In this study, we found that the copy number of *pol* gene could be substantially reduced in both embryonic and cellular levels. Deep-sequencing results showed that about 85% copies of *pol* genes were converted from C to T, indicating that CBE could be used to generate PERV-inactivated pigs.

Off-targeting issue has been a safety concern for gene editing. Recent reports showed that BE3, but not ABE, could induce genome-wide de novo single-nucleotide variants (SNVs) in mice[34] and rice[35]. We only detected one off-target mutation (OT3 for *LMNA*-sgRNA) of 53 POTs (7 for *LMNA*-sgRNA, 21 for *DMD*-sgRNA, 7 for *RAG1*-sgRNA, 8 for *RAG2*-sgRNA, and 10 for *IL2RG*-sgRNA) with PCR followed by Sanger sequencing. The status of genome-wide C-to-T type of SNVs beyond POTs of base-edited pigs achieved in this reports was not verified by whole-genome sequencing. Given that most of the survived base-edited pigs are growing normally, we speculate that the de novo SNVs may not cause fatal phenotypes.

In summary, the CBE system (BE3 and hA3A-BE3) can be used to generate multiplex point mutations efficiently at the cellular and embryonic levels simultaneously. Single and multiple gene point mutations could be introduced into large animals, such as pigs, through embryo injection or nuclear transfer approach. The application of the CBE system should promote the generation of animal models harboring various point mutations or for gene therapy of genetic diseases with multiplex point mutations.

## Methods

**Animals.** Bama miniature pigs and Large White pigs were maintained at the large animal facility of Guangzhou Institute of Biomedicine and Health, Chinese Academy of Sciences. The protocols for the use of pigs were approved by the Institutional Animal Care and Use Committees at Guangzhou Institute of Biomedicine and Health Chinese Academy of Sciences (Animal Welfare Assurance #A5748-01).

**Vector construction.** BE3 fragments were synthesized by Guangzhou IGE Biotechnology Ltd. The fragments were then cloned into the pCS2 vector with the SP6 promoter and into the pCDNA3.1 (+) vector with the CMV promoter. The obtained vectors were named as pCS2-BE3 and pCDNA3.1-BE3. pCMV-hA3A-BE3 vector (#113410) and U6-sgRNA cloning vector (#48962) were purchased from Addgene. pCS2-hA3A-BE3 vector was constructed by ligating linearized pCS2 vector and hA3A-BE3 fragment. *DMD*-sgRNA, *TYR*-sgRNA, *LMNA*-sgRNA, *pol*-sgRNA, *RAG1*-sgRNA, *RAG2*-sgRNA, and *IL2RG*-sgRNA were designed following the G-N19-NGG rule. Two complementary oligonucleotides of sgRNAs were synthesized and then annealed to double-stranded DNAs. The annealed products were then cloned into the BbsI-digested U6-sgRNA cloning vector and obtained sgRNA-expressing plasmid. The primers used above are listed in Supplementary Data 3.

**Production of mRNA and sgRNA.** The pCS2-BE3 and pCS2-hA3A-BE3 vectors were linearized by the restriction endonuclease Not I, which were used as the BE3 and hA3A-BE3 mRNA template. The BE3 and hA3A-BE3 mRNAs were synthesized using mMESSAGE mMACHINE™ SP6 Transcription Kit (Thermo Fisher Scientific, AM1340) and purified with RNeasy MiniElute Cleanup kit (Qiagen) in accordance with the manufacturer's instructions. Templates for sgRNA in vitro transcription were amplified from the constructed U6-sgRNA cloning vectors and then transcribed using HiScribe™ T7 Quick High Yield RNA Synthesis Kit (New England Biolabs). The in vitro-transcribed sgRNAs were purified with miRNeasy Mini Kit (Qiagen) following the manufacturer's instructions. The primers used for amplifying sgRNAs are listed in Supplementary Data 3.

**Pig embryo injection.** For pigs, zygotes from Large White pigs were collected after insemination and transferred into manipulation medium. Given that the porcine oocytes are easily available from a local slaughterhouse, in vitro-activated porcine parthenogenetic embryos were used to evaluate the editing efficiency and viability of pig embryos after injection RNA of the CBE system. Porcine oocyte collection, in vitro maturation, and parthenogenetic activation were conducted as reported in our previous studies[43,44]. In brief, a mixture of BE3 or hA3A-BE3 mRNAs (150 ng/µL) and sgRNAs (50 ng/µL for each sgRNA) was microinjected into the cytoplasm of one-cell stage porcine zygotes and porcine PA embryos. After 6-day in vitro culture, blastocysts of pigs were collected for genotyping by PCR and Sanger sequencing.

**PFFs culture and transfection.** PFFs were isolated from 35-day-old fetuses of Bama miniature pigs. After removing heads, tails, limbs and viscera, the remaining fetuses were digested by 0.5 mg/mL collagenase IV for 2.5 h at 38 ℃. Isolated PFFs were cultured in the PFF culture medium (Dulbecco's modified Eagle's medium (DMEM, HyClone) supplemented with 15% fetal bovine serum (FBS, Gibco), 1% nonessential amino acids (NEAA, Gibco), 2 mM GlutaMAX (Gibco), 1 mM sodium pyruvate (Gibco), and 2% penicillin–streptomycin (HyClone)) for 12 h and then frozen in cell freezing medium (90% FBS and 10% dimethyl sulfoxide).

A day before transfection, PFFs were thawed and cultured in 10 -cm dishes to subconfluence. Then, ~1 × 10^6 PFFs were electroporated with pCDNA3.1-BE3 or pCMV-hA3A-BE3 (10 µg), and sgRNA-expressing vectors (3 µg of each sgRNA) at 1350 V, 30 ms, 1 pulse using the Neon™ transfection system (Life technology). The electroporated cells were recovered for 24 h, and then divided into 20 10 -cm culture dishes. After 10-day G418 selection, individual cell colonies were selected and cultured in 24-well plates. When the confluence reached 90%, the cell colonies were sub-cultured in 12-well plates, and 20% of each cell colony was collected and lysed individually in 10 µL of lysis buffer (0.45% NP-40 plus 0.6% Proteinase K) for 60 min at 56 ℃, and then for 10 min at 95 ℃. The lysate was then used as a template for PCR. The corresponding primers are listed in Supplementary Data 3. The PCR products were further sequenced to identify point mutations. Some PCR products were subjected to sub-clone into the pMD18-T vector (Takara) and then sequenced to determine the exact point mutation patterns and mutation efficiency. Cell colonies with expected mutations were expanded and then cryopreserved in cell freezing medium (90% FBS and 10% dimethyl sulfoxide) for future use.

**SCNT and production of mutant pigs.** Before SCNT, the cryopreserved cell colonies were thawed and cultured in a 24-well plate. Porcine oocyte collection, in vitro maturation, and SCNT were performed as our previous reports[43,44]. Briefly, the selected cumulus oocyte complexes were cultured in maturation medium for 42–44 h at 39 ℃ in an incubator containing 5% CO₂. Matured oocytes were then enucleated and donor cells were injected into the perivitelline space of the oocytes. After injection, fusion and activation were performed using an electrofusion instrument in accordance with the manufacturer's instructions. The reconstructed embryos were then cultured overnight and surgically transferred to the oviducts of the surrogates. After 114 days of gestation, the cloned piglets were delivered through natural birth.

**Genomic DNA extraction and genotyping**. Genomic DNAs were extracted from the ear tissues of the newborn animals by using TIANamp Genomic DNA Kit (TIANGEN) and then subjected to targeted Sanger sequencing and deep sequencing. The primers used for animal genotyping were the same as those used for cell- and embryo-level genotyping.

**H&E staining**. The thymus and spleen tissues obtained from WT and *RAG1*, *RAG2*, and *IL2RG* mutant pigs were fixed in 4% paraformaldehyde for 2 days. The fixed tissues were embedded in paraffin wax and cross-sectioned at 3 μM using a vibratome for H&E staining. The slides were deparaffinized with xylene and subsequently rehydrated with 100%, 90%, 80%, 70%, and 50% alcohol, followed by $H_2O$. Finally, the rehydrated slides were stained with H&E and viewed under a fluorescence inversion microscope.

**RT-PCR and qRT-PCR analysis**. The total RNAs were extracted from the ear tissues of WT and $LMNA^{G608G}$ piglets, and the thymus and spleens of WT and *RAG1*, *RAG2*, and *IL2RG* mutant piglets by using TRIzol Reagent (Life technologies) in accordance with the supplier's manual. The total RNAs (5 μg) were used to synthesize cDNAs by using ReverTra Ace (Toyobo) and oligo-dT (Takara). For *RAG1*, *RAG2*, and *IL2RG* mutant piglets, transcript levels of *CD4*, *CD8*, *CD19*, and *IL2RG* were evaluated by qRT-PCR using the CFX96 real-time system (BIO-RAD) with SYBR Green Supermix (172-5274, BIO-RAD). For $LMNA^{G608G}$ piglets, RT-PCR was performed to detect the expression of progerin. The primers used for RT-PCR and qRT-PCR are listed in Supplementary Data 3.

**Flow-cytometry analysis**. Peripheral blood mononuclear cells were isolated from whole blood of *RAG1*, *RAG2*, and *IL2RG* mutant piglets and age-matched control piglets. Red blood cells were eliminated by using ACK lysis buffer (Leagene). To identify porcine $CD3^+$ T, $IgM^+$ B, and $CD3^-CD8^+$ NK cells, we used mouse anti-pig CD3 (BD, 561476, 1:200), CD8 (BD, 559584, 1:200), and goat anti-pig IgM (AbD, AAI39F, 1:200). Samples were analyzed using an Accuri C6 flow cytometer (Accuri Cytometers, Ann Arbor, MI).

**PCR analysis to identify V(D)J recombination**. DNA samples were extracted from the peripheral blood, spleen, thymus and born marrow of RAG1, RAG2, and IL2RG mutant piglets and age-matched control piglets. The primers for identifying V(D)J recombination are shown in Supplementary Data 3. TCR-β recombination and IgH genes were analyzed by PCR using 2 × SuperStar PCR Mix (GenStar). We used the D1J1-F, D1J1-R, FR1, and JF primers to identify the rearranged of TCR-β and IgH. The thermal cycler parameters were as follows: 95 ℃ for 5 min; 35 cycles of 98 ℃ for 15 s, 68 ℃ for 30 s, and 72 ℃ for 2 min; and a final step at 72 ℃ for 5 min.

**Off-target analysis**. The POT sites for each sgRNA were predicted to analyze site-specific edits by the BE3 system according to an online design tool (http://www.rgenome.net/cas-offinder/)[33] allowing for un-gapped alignment with up to three mismatches in the sgRNA target sequence. All POTs were amplified by PCR and then subjected to Sanger sequencing to confirm the off-target effects, respectively. The primers for amplifying the off-target sites are listed in Supplementary Data 3.

**Reporting summary**. Further information on research design is available in the Nature Research Reporting Summary linked to this article.

## Data availability

Previously constructed plasmids are available from Addgene under ID numbers 113410 and 48962. All deep-sequencing data from this study has been deposited at NCBI Sequence Read Archive (SRA) under accession numbers SRR8776264, SRR8776265, and SRR8776266. The source data underlying Figs 3f, 3h, and 6a, 6e–i are provided as a Source Data file. All relevant data supporting the findings of this study are represented within the article or available from the corresponding authors upon reasonable request.

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

## Acknowledgements

This work was financially supported from the Strategic Priority Research Program of the Chinese Academy of Sciences (XDA16030503, XDA16030501), the National Key Research and Development Program of China Stem Cell and Translational Research (2017YFA0105103), Key Research & Development Program of Guangzhou Regenerative Medicine and Health Guangdong Laboratory (2018GZR110104004), the National Natural Science Foundation of China (81702115, 81672317, 31871292), the Youth Innovation Promotion Association of the Chinese Academy of Sciences (2019347), Science and Technology Planning Project of Guangdong Province, China (2014B020225003, 2016A030313169, 2017B030314056, 2017A050501059), the Bureau of Science and Technology of Guangzhou Municipality (201704030034), and the Science and Technology Planning Project of Jiangmen (2017TD02).

## Author contributions

L.L. and K.W. designed the study. J.X., W.G., N.L., and Q.L. performed experiments and analyzed the data. W.G., F.C., Z.O., Q.Z., C.L., and N.F. performed SCNT and porcine embryos injection. Y.Z. and Z.L. performed embryos transfer experiments and provided animal (pig) care. S.G. performed bioinformatics analysis. X.Y., X.H., H.W., Q.J., H.S., Y.L., T.L., and L.Q. provided technical assistance. X.L. provided valuable comments. K.W. and L.L. supervised the project and wrote the paper.

## Additional information

**Competing interests:** The authors declare no competing interests.

