## [Peer Review File · Nature Communications]

Reviewers' Comments:

Reviewer #1:

Remarks to the Author:

Xie et al. used base editing technology to induce multiple targeted C-to-T base substitutions in porcine embryos and porcine somatic cells, and then they generated base-edited pigs by transferring the nuclear of edited somatic cells. The authors also generated multiple-base-edited rabbits with zygote injection. This study is an important extension of base editing application as it clearly shows that base editors can induce targeted multiple-site base substitutions in big animals such as pigs and rabbits. However, there are still a few concerns about this study.

Major concerns

The authors used somatic cell nuclear transfer to generate base-edited pigs. However, they generated base-edited rabbits conveniently with one-step zygote injection. Thus, it is important to know whether the authors can generate base-edited pigs with one-step zygote injection.

Recently, human APOBEC3A (A3A)-derived base editors (A3A-BEs) have been developed to induce efficient base editing in mammalian cells and plants (Wang et al., Nat Biotech, 2018; Zong et al., Nat Biotech, 2018). Thus, it is worthy to compare the editing efficiencies of BE3 and A3A-BEs in porcine embryos and in rabbit zygotes.

The English writing needs to be substantially improved.

Minor concerns

It is better to use the title "Efficient base editing..." instead of "Highly efficient base editing..." for this manuscript.

Reviewer #2:

Remarks to the Author:

The submitted manuscript, "Highly efficient 1 base editing for multiple genes/loci in rabbits and pigs using base editors" by Xie et al., describes based editing in multiple loci in rabbit and pig models. The manuscript described interesting findings, but was found to be rather unfocused and underdeveloped and was recommended for major revision across multiple fronts. Major and minor concerns are listed below:

1) Base editing in rabbit models have recently been reported by the same group in Nature Communications. In effect, this manuscript is a continuation of the work. Based on the recent publication, it is not clear editing in more than one gene should qualify for publication in this prestigious journal.

2) Base editing in pigs is of interest to the community, but I think the manuscript would have been more valuable had it focussed on the pig model. For one, authors described base editing in parthenogenic embryos, but did not generate offspring from direct zygotic injections. But instead, they proceeded with edited in the somatic cells for generation of offspring by SCNT/cloning. And in the end they included data from rabbits to bulk up the manuscript.

3) The manuscript would have been benefitted by focussing on embryo injections and somatic cell editing in pig models across the target genes and comparing the relative efficiencies in each model

In summary, the authors should be advised to focus on pig studies. Generating pregnancies by

embryo injections as was done with the rabbit models would significantly improve the impact of the manuscript.

The authors should also focus on improving the discussion section. There is a need to expand discussion, especially lack of efficiency or specificity at target sites, range finding experiments, and deep sequencing (as was done with the rabbit models) in pig models to look at on-and off target sites. Specifically, the authors need to discuss/explore further the unwanted C-to-A substitutions and 18bp deletions.

Based on these, I recommend rejecting the manuscript and requesting re submission with a more focused revised article focusing on pig models.

Dear Editors and Reviewers:

Thank you very much for sending our work (NCOMMS-18-32570) for peer review and inviting us to revise our work. We appreciate the positive and encouraging comments from the reviewers. Those comments are all valuable and very helpful for revising and improving our paper, as well as the important guiding significance to our researches. We have studied comments carefully and performed additional experiments to address the questions. The discussion section was completely rewritten as reviewer suggested. We revised the manuscript in response to the reviewer comments. Our point-by-point responses are presented below.

Reviewers' comments:**Reviewer #1 (Remarks to the Author):**

Xie et al. used base editing technology to induce multiple targeted C-to-T base substitutions in porcine embryos and porcine somatic cells, and then they generated base-edited pigs by transferring the nuclear of edited somatic cells. The authors also generated multiple-base-edited rabbits with zygote injection. This study is an important extension of base editing application as it clearly shows that base editors can induce targeted multiple-site base substitutions in big animals such as pigs and rabbits. However, there are still a few concerns about this study.

Response: We very appreciated for the reviewer's positive comments and advices.

Major concerns

The authors used somatic cell nuclear transfer to generate base-edited pigs. However, they generated base-edited rabbits conveniently with one-step zygote injection. Thus,

it is important to know whether the authors can generate base-edited pigs with.

Response: We appreciate the reviewer's important suggestion for improving our manuscript. We performed supplementary experiments to generate base editing pigs by embryo injection. Eight Hutchinson-Gilford progeria syndrome (HGPS) piglets (1 stillborn and 7 liveborn) carrying a c.1824C>T (pG608G) point mutation were achieved via one-step zygote injection of BE3 mRNA and *LMNA*-sgRNA. Consistent with human HGPS patients, the mutant pigs could express progerin mRNA and protein. We added following paragraphs in Results section.

“Generation of *LMNA*^{G608G} pig via zygote injection

Although single-cell colonies harboring desired *LMNA*^{G608G} mutation were effectively obtained (41.7%, 43/103) when using BE3 and *LMNA*-sgRNA (**Fig. 2c**), these colonies exhibited premature senescence phenotypes and were not suitable used as the nuclear donor for SCNT (**Supplementary Fig. 7**). Therefore, we next investigated whether *LMNA*^{G608G} mutation pig model could be generated by using embryo injection of base editors. *In vitro* transcribed BE3 mRNA and *LMNA*-sgRNA were co-injected into the cytoplasm of one-cell stage zygotes from for Large White sows. Forty-six injected porcine zygotes were then transferred into two surrogate Large White sows. A total of nine piglets (5 male and 4 female, 1 stillborn and 8 liveborn) were born at term from one pregnant surrogate (**Fig. 3a-b**). Genotype analysis showed that 8 piglets (88.9%, 8/9) harbored C-to-T mutation (**Fig. 3c-d, Supplementary Fig. 8**). For 357-2, 357-3, 357-4 and 357-7 (44.4%, 4/9), undesired indels (1bp or 12bp deletion) were also found at target sites with the efficiency of 44%, 15%, 3%, and

28%, respectively (**Fig. 3c, Supplementary Fig. 8**). Targeted deep sequencing results showed that 13-100% site-specific C>T mutations were achieved at *LMNA* c.1824C site for piglet 357-1, 357-2, 357-3, 357-5, 357-6, 357-7, 357-8, and 357-9 (**Fig. 3c-e**). Notably, piglet 357-8 harbored homozygous c.1824C > T mutations, but unfortunately died within 2 days (**Fig. 3c-e**). The heart, liver, spleen, lung, kidney of piglet 357-8 were collected and Sanger Sequencing results showed that homozygous c.1824C > T mutations were observed in all these tissues (**Supplementary Fig. 9**). These results showed that pig models carrying C-to-T substitutions can efficiently generated by direct zygotes injection of BE3. Sanger sequencing analysis of 7 POT sites showed that 1 off-target mutation (OT3) was found in 8 (88.9%, 8/9) base-edited piglets (**Supplementary Fig. 10**), consistent with recent reports for cytosine base editors can induce genome-wide off-target mutations in mammal³³ and plant³⁴.

To test whether *LMNA*^{G608G} mutation could cause aberrant mRNA splicing, total RNAs from the ear tissues were extracted. RT-PCR and Sanger Sequencing analysis showed that ear tissues from all *LMNA*^{G608G} piglets expressed a smaller mRNA with a 150-nucleotide deletion (**Fig. 3f-g**). Western Blot analysis of the heart, liver, spleen, lung, kidney tissues of piglet 357-8 demonstrated that progerin, a truncated splicing mutant of lamin A, was present in the tissues of *LMNA*^{G608G} mutation piglets but not in wide-type tissues (**Fig. 3h**). Taken together, these results demonstrated that *LMNA*^{G608G} mutation piglets, consistent with human HGPS patients, can express progerin mRNA and protein.

Recently, human APOBEC3A (A3A)-derived base editors (A3A-BEs) have been

developed to induce efficient base editing in mammalian cells and plants (Wang et al., Nat Biotech, 2018; Zong et al., Nat Biotech, 2018). Thus, it is worthy to compare the editing efficiencies of BE3 and A3A-BEs in porcine embryos and in rabbit zygotes.

Response: We very appreciated for the reviewer's value advices. We performed supplementary experiments to compare the editing efficiencies of BE3 and hA3A-BE3 in porcine embryos and somatic cells as reviewer suggested. To address the question of reviewers in detail, we added following sentences in Results section.

“Recently, a new member of cytosine base editors, hA3A-BE3, has been harnessed for base editing the genomes of mammalian cells and plants with improved base editing efficiency and specificity²⁸⁻³⁰. We also tested multiple base editing capacity of hA3A-BE3 in pig embryos. As in BE3, DTL and R12I were used for evaluation of hA3A-BE3 system in porcine embryos. As shown in **Supplementary Fig. 2a-e**, 50% embryos for DTL (5/10) and 41.7% embryos for R12I (5/12) harbored C>T mutations at target sites of all three genes, which were comparable to those of the BE3 system (50% for DTL and 36.4% for R12I).”

“We also tested multiple base editing capacity of hA3A-BE3 in porcine cell level. As in BE3, DTL and R12I were used for evaluation of hA3A-BE3 in porcine fibroblasts. Base editing at target sites of all three genes occurred in the genomes of 55% single cell colonies for DTL (11/20) and 40% for R12I (8/40), which was higher than BE3-mediated multiplex base editing (25.2% for DTL and 1.5% for R12I) (**Supplementary Fig. 5a-e**).”

The English writing needs to be substantially improved.

Response: Thanks for reviewer's suggestion. We have sent the manuscript to English editing company to polish the English writing for the revised version.

Minor concerns

It is better to use the title "Efficient base editing..." instead of "Highly efficient base editing..." for this manuscript.

Response: Thanks for reviewer's suggestion. We revised the title as "**Efficient base editing for multiple genes/loci in pigs using base editors**" in the revised manuscript.

Reviewer #2 (Remarks to the Author):

The submitted manuscript, "Highly efficient 1 base editing for multiple genes/loci in rabbits and pigs using base editors" by Xie et al., describes based editing in multiple loci in rabbit and pig models. The manuscript described interesting findings, but was found to be rather unfocused and underdeveloped and was recommended for major revision across multiple fronts.

Major and minor concerns are listed below:

1) Base editing in rabbit models have recently been reported by the same group in Nature Communications. In effect, this manuscript is a continuation of the work. Based on the recent publication, it is not clear editing in more than one gene should qualify for publication in this prestigious journal.

Response: Actually, we did publish a paper in *Nature Communications* in last year named "**Highly efficient RNA-guided base editing in rabbit.**" Although rabbit is a classic animal model species, pig is more similar to humans than rabbit in organ size, anatomy, physiology, metabolism and immunology. Thus, pigs are considered as ideal

animal models for xenotransplantation, bioreactor and human diseases. Many traits in agriculture are caused by multiple SNPs, and many genetic diseases in biomedicine arise from point mutations in multiple sites. Therefore, base editing of the genome in multiple sites is necessary to achieve favorable traits in agriculture, establish human disease animal models, and to treat human hereditary diseases. We believe that our findings and data are suitable for publication in *Nature Communications* and of broad interest to *Nature Communications* readers.

2) Base editing in pigs is of interest to the community, but I think the manuscript would have been more valuable had it focussed on the pig model. For one, authors described base editing in parthenogenic embryos, but did not generate offspring from direct zygotic injections. But instead, they proceeded with edited in the somatic cells for generation of offspring by SCNT/cloning. And in the end they included data from rabbits to bulk up the manuscript.

Response: Thanks for reviewer's suggestion. Yes, we used parthenogenetically activated oocytes, which are easily accessible from slaughter house, to test the efficiency of CBEs system. To address the reviewer's comments, we performed supplementary experiments to generate base editing pigs by embryo injection approach. Eight Hutchinson-Gilford progeria syndrome (HGPS) piglets (1 stillborn and 7 liveborn) carrying a c.1824C > T (pG608G) point mutation have been achieved via one-step zygote injection of BE3 mRNA and *LMNA*-sgRNA. Consistent with human HGPS patients, the mutant pigs could express progerin mRNA and protein. We added these results in the Results section. In addition, to focus the work on pigs as the

reviewer suggested, we removed the data of rabbits in the Results section of revised manuscript.

3) The manuscript would have been benefitted by focusing on embryo injections and somatic cell editing in pig models across the target genes and comparing the relative efficiencies in each model

Response: Thanks for reviewer's suggestion. In our original manuscript, we did present data about the efficiency of CBE system in both embryo level and cell level, which are corresponding to the efficiency of generation mutant pigs by embryo injection and SCNT, respectively. In the section of discussion of revised manuscript, we compared possible practical applications between embryo injection and SCNT. In Discussion section, we added following paragraph to explain how to choose SCNT or direct embryo injection to generate gene-modified pig models:

“When using embryo injection to generate gene editing animals, the event of C-to-T conversion may happen in a cell of two or more cell stage embryos. Thus, many of the resulted founder animals were chimeric ones mixed with mutant tissues and non-mutant tissues, or mixed with homozygous cells and heterozygous cells. To acquire the animals with a pure pattern of mutation, one or two more rounds of further breeding have to be employed for selection among the offspring, which is expensive and time-consuming. For multiple base editing, occurrence of the undesired mutations in founders would make the chimeric issues more complicated. Therefore, it is not applicable using embryo injection to generate multiple base editing large animals such as pigs with long gestation term and sex maturation time. However, when generating

a single base editing animal, for some genes with special functions, for instance, *LMNA* gene, an aging related gene, embryo injection could be a better choice. Mutations of *LMNA* gene could cause premature aging syndromes. When we putatively established porcine fibroblast line harboring *LMNA*^{G608G} mutation, the cells also exhibited premature senescence phenotypes *in vitro*, and were not suitable to be used as the nuclear donors for SCNT. Therefore, we had to use embryo injection of base editors to make *LMNA*^{G608G} mutation pig model. HGPS pig models carrying a c.1824C>T (pG608G) point mutation were successfully generated via zygotes injection, indicating that pig models carrying C-to-T substitutions can be efficiently generated by direct zygotes injection of BE3 and provided an appropriate model for aging preclinical study.

Gene-targeting large animals can be made through combination of gene targeting of somatic cells with somatic cell nuclear transfer^{1,2,38}. *In vitro* screening of gene-edited fibroblasts with desired mutations prior to SCNT could circumvent the problems such as chimerism and unwanted mutations, which were inevitably found in embryo injection approach. For this approach, since the pig cloning technology has been well established, the most elusive issue for generation of gene editing animal is to create mutant cells used as donor nuclei. Therefore, we tested the multiple base editing efficiency of BE3 and hA3A-BE3 in porcine somatic cells. Our results showed that, both BE3 and hA3A-BE3 could efficiently create base editing in three genes simultaneously. Not as that in embryos level, in which hA3A-BE3 and BE3 had similar efficiency, the efficiency of hA3A-BE3-mediated multiplex base editing was

higher than that of BE3 in porcine cell level. Although as in the embryo level, BE3 system also could result in cells with one or two gene mutations, and unwanted mutations such as undesired non-C-to-T substitutions and indels in the some cell colonies, we could choose the cell lines with desired mutations based on genotype test to make cloned pigs with favorable phenotypes. The effectiveness of this approach was validated by the generation base editing pigs with mutation of one gene and three genes, respectively. We first made a DMD pig model with C-to-T conversion of a single gene (*DMD* gene) through BE3 system combining with SCNT approach. DMD is an X-linked recessive hereditary disease and the average life expectancy of patients with this disorder is 26 years old³⁹. Previously reported *DMD*^{-/-} or *DMD*^{-Y} pigs made by deletion of a fragment through CRISPR/Cpf1 system could not survive beyond 3 months¹. Thus, we chose female cell colonies with *DMD*^{+/-} mutation of C-to-T conversion to generate *DMD* heterozygous female pigs, which could survive more than one year and could be used to mate with *DMD*^{+Y} pigs to achieve many F1 pig models with DMD phenotypes for biomedicine study. For validate the application of BE3 system in generation of multiple base editing animal, by choosing the mutant cells with C-to-T conversion of three functionally related genes *RAG1*, *RAG2* and *IL2RG* selected to perform SCNT, we successfully achieved immunodeficient pigs lack of B cells, T cells and NK cells, which is consistent with the SCID pigs made by deletion of fragments with other gene editing technologies⁴⁰⁻⁴².”

In summary, the authors should be advised to focus on pig studies. Generating pregnancies by embryo injections as was done with the rabbit models would

significantly improve the impact of the manuscript.

Response: Yes, as the reviewer suggested, we focused our work on pigs and removed contents about rabbits.

The authors should also focus on improving the discussion section. There is a need to expand discussion, especially lack of efficiency or specificity at target sites, range finding experiments, and deep sequencing (as was done with the rabbit models) in pig models to look at on-and off target sites. Specifically, the authors need to discuss/explore further the unwanted C-to-A substitutions and 18bp deletions.

Response: We have completely rewrote the discussion section in revised manuscript according to reviewer's suggestion, including efficiency and specificity at target sites, and the unwanted C-to-A substitutions and undesired deletions. For the question of deep sequencing (as was done with the rabbit models) in pig models, because of SCID and DMD pig models were generated from a single cell, in which the genotypes had been analyzed before somatic cell nuclear transfer, mutation patterns of these pigs were pure and the deep sequencing is not necessary on animal level. But for HGPS model generated by embryo injection, targeted deep sequencing was performed to analyze the mutation pattern of base editing.

We are grateful for your favorable consideration of our manuscript. We hope that the revised manuscript submitted is ready for publication.

Sincerely,

Liangxue Lai, Ph. D.

Reviewers' Comments:

Reviewer #1:

Remarks to the Author:

The manuscript has been improved substantially and all the concerns have been satisfactorily addressed. I suggest to accept this revised manuscript.

Reviewer #2:

Remarks to the Author:

The revised manuscript is substantially improved. The authors' addressed all major concerns of the reviewer and is therefore recommended for publication. The authors are encouraged to address these minor concerns in their final submission for publication.

The authors are encouraged to included a brief discussion on the recent publication (Cytosine base editor generates substantial off-target single-nucleotide variants in mouse embryos, by ZOU et al) and put their work into perspective.

The authors are encouraged to revise the manuscript for english language and editorial needs.

We appreciate the positive feedback on this manuscript. The point-by-point responses of reviewers and editors are listed below.

Responses to the Reviewers' Comments

Reviewer #1 (Remarks to the Author):

The manuscript has been improved substantially and all the concerns have been satisfactorily addressed. I suggest to accept this revised manuscript.

Response: Thanks to the positive comments of our revised manuscript.

Reviewer #2 (Remarks to the Author):

The revised manuscript is substantially improved. The authors' addressed all major concerns of the reviewer and is therefore recommended for publication. The authors are encouraged to address these minor concerns in their final submission for publication.

Response: We very appreciated for the reviewer's positive comments and advices.

The authors are encouraged to include a brief discussion on the recent publication (Cytosine base editor generates substantial off-target single-nucleotide variants in mouse embryos, by Zuo et al) and put their work into perspective.

Response: Thanks for reviewer's suggestion. To discuss the off-target effects of Cytosine base editor and put the work of Zuo et al into perspective, we added following sentences in Discussion section.

“Off-targeting issue has been a safety concern for gene editing. Recent reports showed that BE3, but not ABE, could induce genome-wide de novo single-nucleotide variants (SNVs) in mice³⁴ and rice³⁵. We only detected one off-target mutation (OT3 for *LMNA*-sgRNA) of 53 POTs (7 for *LMNA*-sgRNA, 21 for *DMD*-sgRNA, 7 for *RAG1*-sgRNA, 8 for *RAG2*-sgRNA, and 10 for *IL2RG*-sgRNA) with PCR followed by Sanger sequencing. The status of genome-wide C-to-T type of SNVs beyond POTs of base-edited pigs achieved in this reports was not verified by whole genome sequencing. Given that most of the survived base-edited pigs are growing normally, we speculate that the de novo SNVs may not cause fatal phenotypes.”

The authors are encouraged to revise the manuscript for English language and editorial needs.

Response: We have revised the manuscript in accordance with the editorial requests.